Metabolome and transcriptome integration reveals insights into petals coloration mechanism of three species in Sect. Chrysantha chang

Yan Yadan 1
Wang Ye 1
Wen Yafeng 1 wenyafeng7107@163.com
Huang Yu 2 huangyu@nnxy.edu.cn
Zhang Minhuan 1
Huang Jiadi 1
Li Xinyu 1
Wang Chuncheng 1
Xu Dangqing 2
1 College of Landscape Architecture, Central South University of Forestry and Technology , Changsha , China
2 Nanning University , Nanning , China
Irfan Mohammad
Electronic publication date: 2024 Apr 19
Publication date: 2024
Volume: 12
Electronic Location ID: e17275
Received 2023 Sep 7; Accepted 2024 Apr 1
Copyright: © 2024 Yan et al.
Copyright year: 2024
Copyright holder: Yan et al.
License: This is an open access article distributed under the terms of the Creative Commons Attribution License, which permits unrestricted use, distribution, reproduction and adaptation in any medium and for any purpose provided that it is properly attributed. For attribution, the original author(s), title, publication source (PeerJ) and either DOI or URL of the article must be cited.
License URL: https://creativecommons.org/licenses/by/4.0/

Keywords: Sect. Chrysantha chang, Flower color, Flavonoid biosynthesis, Candidate genes, Transcription factors

Funding: The authors received no funding for this work.

==============================
Background

Sect. Chrysantha Chang, belonging to the Camellia genus, is one of the rare and precious ornamental plants distinguished by a distinctive array of yellow-toned petals. However, the variation mechanisms of petal color in Sect. Chrysantha Chang remains largely unclear.

Methods

We conducted an integrated analysis of metabolome and transcriptome to reveal petal coloration mechanism in three species, which have different yellow tones petals, including C. chuongtsoensis (CZ, golden yellow), C. achrysantha (ZD, light yellow), and C. parvipetala (XB, milk white).

Results

A total of 356 flavonoid metabolites were detected, and 295 differential metabolites were screened. The contents of 74 differential metabolites showed an upward trend and 19 metabolites showed a downward trend, among which 11 metabolites were annotated to the KEGG pathway database. We speculated that 10 metabolites were closely related to the deepening of the yellowness. Transcriptome analysis indicated that there were 2,948, 14,018 and 13,366 differentially expressed genes (DEGs) between CZ vs. ZD, CZ vs. XB and ZD vs. XB, respectively. Six key structural genes (CcCHI, CcFLS, CcDFR1, CcDFR2, CcDFR3, and CcCYP75B1) and five candidate transcription factors (MYB22, MYB28, MYB17, EREBP9, and EREBP13) were involved in the regulation of flavonoid metabolites. The findings indicate that flavonoid compounds influence the color intensity of yellow-toned petals in Sect. Chrysantha Chang. Our results provide a new perspective on the molecular mechanisms underlying flower color variation and present potential candidate genes for Camellia breeding.

Introduction

The evaluation of ornamental and economic significance in ornamental plants is heavily reliant on the characterization of flower color. Recent studies, as highlighted by Qiao et al. (2022), underscore the pivotal role of primary pigment categories such as flavonoids, carotenoids, and alkaloids, as well as their respective quantities, in determining flower coloration. Flavonoids, as a diverse group of secondary metabolites, contribute to a wide spectrum of hues to the majority of plants species, as outlined by Li et al. (2022). Up to now, the flavonoid biosynthesis pathway has been elucidated as a relatively clear metabolic route (Zhu et al., 2021). Originating from phenylalanine, this pathway undergoes a sequence of enzymatic reactions to generate a range of metabolites (Nabavi et al., 2020). The synthesis of flavonoids involves two essential types of genes: structural genes and transcription factors (TFs). The former genes encode the enzymes directly involved in the biosynthesis of flavonoids, whereas the latter genes control the expression of these structural genes (Li et al., 2017). The structural genes have garnered considerable attention and wield substantial influence over the modification of flower color (Yang et al., 2018). An illustration of this is the collective expression of THC2’GT, CHI, and FNS II, which contributes to the observed yellow pigmentation in Paeonia delavayi (Shi et al., 2015). In P. lactiflora, a different scenario emerges where PlPAL, PlFLS, PlDFR, PlANS, Pl3GT, and Pl5GT collectively suppress anthocyanin biosynthesis, thereby orchestrating the formation of yellow pigmentation (Zhao et al., 2014). Moreover, the regulatory mechanism of three extensively researched TFs families, namely MYB, bHLH, and WD40-demonstrate a relatively clear regulatory mechanism (Feng et al., 2020). Recent studies have explored additional TF families associated with flavonoid synthesis in ornamental plants, including AP2/EREBP, WRKY, MADS-Box, and bZIP families in P. suffruticosa, Phalaenopsis amabilis and Brassica rapa (Zhao et al., 2015; Meng et al., 2020; Rameneni et al., 2020). However, these studies suggested that the molecular mechanism dictating color variations among distinct plant species lack uniformity.

The global Camellia genus comprises over 300 species encompassing nearly 30,000 varieties. Among them, several species are deliberately cultivated for their ornamental qualities, characterized by lustrous evergreen foliage and striking winter or spring blossoms (Li et al., 2021). Predominantly, Camellia flowers exhibit colors in shades of red, white, or compound hues, with yellow flowers being relatively scarce (He et al., 2018). Sect. Chrysantha Chang represents a rare and coveted genetic germplasm resource crucial for cultivating yellow Camellia, exhibiting a diverse spectrum of yellow tones petals (Gao et al., 2014; Tang et al., 2006). Currently, Zhou et al. (2017) have elucidated the co-expressed network responsible for the carotenoid pathway of C. nitidissima. Liu et al. (2023) elucidated the regulatory mechanism governing flavonoid synthesis within the petals of C. nitidissima. However, their studies were limited to a few selected species. Considering the intricate nature of flavonoid biosynthesis and the nuanced interplay of diverse enzymes shaping plant coloration, a significant gap exists in comprehending the metabolic and transcriptional variances among the distinct species of Sect. Chrysantha Chang, which manifest varying degrees of yellow flowers. Thus, undertaking species-specific investigations becomes imperative to unravel the precise molecular mechanisms underlying the modulation of flower color in Sect. Chrysantha Chang.

In this study, we analyzed the types and contents of flavonoid metabolites, as well as profiling the association between genes and metabolites. Three goals are identified: (1) to determine the main metabolites enhancing the yellow pigmentation of petals in Sect. Chrysantha Chang; (2) to identify functional structural genes and TFs involved in flavonoid biosynthesis; and (3) to explore the metabolic pathways and the mechanism governing color formation in yellow Camellia petals. The study will provide guidance for the molecular breeding, development, and application of yellow flowers in Camellia.

Materials and Methods

Plant materials

Three species of Sect. Chrysantha Chang, each displaying noticeable differences in their petal yellow phenotype, were selected for investigation, including C. chuongtsoensis (CZ), C. achrysantha (ZD), and C. parvipetala (XB) (Fig. 1A). The flowers of CZ exhibited a golden yellow hue, ZD displayed a light yellow color, and XB presented a milky white appearance when in full bloom. All the materials used were cultivated at Golden Flower Tea Park in Guangxi, China (108°20′53″E, 22°49′11″N). On December 20, 2021, 20 fresh petals at the full-bloom stage were sampled for each repetition. After being rinsed with distilled water to remove surface moisture, three flowers per replicate were used for phenotypic measurements and pigment determination. Additionally, remaining samples were promptly frozen in liquid nitrogen, and subsequently stored at –80 °C for subsequent metabolomic and transcriptomic sequencing analyses. All experiments were conducted in triplicates for each species.

Figure 1 Differences in petal phenotype and pigment content of three species of Sect. Chrysantha Chang.

(A) The materials used in the experiment. (B) The petal color parameters; the L* means lightness, a* and b* means chromatic components, C means chroma, h° means hue angle. (C) The total flavonoid and carotenoid contents. Different letters above bars represent significant difference at P < 0.05. Data represent the means of three biological replicates ± SD.

Phenotypic measurement of petal color

The hue parameter serves as a digital metric to characterize the flower color phenotype.. Hue parameters (L*, a*, b*, C, and h°) of the epidermis of petals and subepidermal layer were measured using the CM-700d colorimeter (Konica Minolta, Japan) to quantify the color of petals across three species (CZ, ZD, and XB). The values of C and h° were calculated according to following formulae: C = (a * 2 + b * 2)1/2 and h° = arctan (b*/a*), respectively (Cui et al., 2019). The parameters of each flower were measured six times, with three flowers per material. The one-way analysis of variance (ANOVA) was conducted using IBM SPSS 23.0 (IBM SPSS Software Inc., Chicago, IL, United States). Statistically significant was tested at the levels of p < 0.01 and p < 0.05.

Measurement of total carotenoid and flavonoid contents

The contents of total carotenoids and flavonoids were measured by visible spectrophotometry employing identical kits (No: BC4330, Beijing Solarbio Science & Technology Co., Ltd, Beijing, China), following the manufacturer’s instructions. Each sample was replicated three times. To determine the total carotenoid content, 0.1 g of fresh petals from each species was mixed with 1 ml of distilled water and 10 mg of reagent 1 (provided in the kit). After thorough grinding, the mixture was transferred to a 10 ml test tube, diluted with an 80% aqueous acetone solution to a final volume of 10 ml, and left to stand for 2 h in darkness. The absorbance of the final extract solution was measured using a spectrophotometer at 440 nm. The total carotenoid content was calculated using the formula: Total carotenoid (mg/g) = 0.04 × A440 × F/W, where F denotes the dilution ratio, and W represents the sample weight (g). The determination of total flavonoid content followed a structured procedure. Initially, approximately 0.1 g of dried petals from each species were pulverized into a powder, combined with 1 ml of a 60% aqueous ethanol solution, and then extracted at 60 °C for 2 h. Subsequently, the resulting extract underwent centrifugation at 12,000 rpm at 25 °C for 10 min. The supernatant was taken and diluted with 60% aqueous acetone solution to a final volume of 1 ml. The resultant sample solution was configured, and the absorbance (ΔA) was measured at 470 nm. The calculation of flavonoid content employed the formula: Flavonoid content (mg/g) = X/W, where X was derived from the standard curve equation: ΔA = 0.5221X + 0.0277 (R2 = 0.9774), and W indicated sample mass (g). To measure the contents of carotenoids and flavonoids, IBM SPSS 23.0 was utilized for conducting ANOVA analysis and Tukey’s honestly significant difference test.

Extraction of metabolome samples for metabolite analysis

The freeze-dried petals, subjected to triple repetition, were comminuted using a mixer mill (MM 400, Retsch) with a zirconia bead for 1.5 min at 30 Hz. Subsequently, the 100 mg of lyophilized powder was dissolved in 1.2 ml 70% methanol solution. The solution underwent vortexed for 30 s per cycle with intervals of 30 min, repeated 6 times. The sample was refrigerated at 4 °C overnight. Following centrifugation at 12,000 rpm for 10 min, the supernatant was collected and filtrated using a filter membrane (SCAA-104, 0.22 μm pore size; ANPEL, Shanghai, China, http://www.anpel.com.cn/). The sample extracts underwent Ultra Performance Liquid Chromatography-Tandem Mass Spectrometry (UPLC-MS/MS) analysis using a UPLC-ESI-MS/MS system (UPLC, SHIMADZU Nexera X2, https://www.shimadzu.com.cn/; MS, Applied Biosystems 4500 Q TRAP (Yu et al. 2020)). Each sample was replicated three times.

The sample quality control (QC) analysis was conducted to assess the stability of the instrument, ensuring the reproducibility and reliability of the data. The multiple reaction monitoring (MRM) was performed to compare the differences in flavonoids and carotenoids (Chen et al., 2013). Utilizing the UPLC-MS/MS detection platform (Metware Biotechnology Co., Ltd., Wuhan, China), qualitative and quantitative analyses were performed using Analyst 1.6.3 (Dong et al., 2022). Principal component analysis (PCA) was performed using “FactoMineR” in R 4.1.3 to characterize the accumulation patterns of metabolites across different samples (Zhang et al., 2022). Differentially accumulated flavonoids (DAFs) were identified based on the threshods of variable importance in the project (VIP) value ≥1 and the fold change (FC) value ≥2 or ≤0.05. All DAFs were annotated using the KEGG compound database (http://www.kegg.jp/kegg/compound/). Subsequently, the annotated metabolites underwent KEGG pathway enrichment analysis (http://www.kegg.jp/kegg/pathway.html), and the significance of metabolite-enriched pathways was determined by the hypergeometric test’s p-value.

RNA-seq library construction and sequencing

Total RNA extraction was performed using freeze-dried petals and the RNA prep Pure Kit DP432 (TIANGEN Biotech Co., Ltd., Beijing, China). The integrity of all RNA samples was assessed evaluated using Qsep1 instrument (Li et al., 2018). A total of 1 μg of total RNA was utilized for constructing RNA libraries with the VAHTS mRNA-seq V3 Library Prep Kit. The library construction and sequencing procedures, which encompassed polyA-selected RNA extraction, RNA fragmentation, random hexamer-primed reverse transcription, and 150 nt paired-end sequencing, were executed using the Illumina HiSeq X-ten platform at Igenebook Biotechnology Co., Ltd, Wuhan, China.

Gene expression level and differential expression analysis

Raw RNA-seq reads were subjected to filtering procedures aimed at removing low-quality reads, characterized by the presence of more than 50% bases with Q-value ≤ 20, as well as reads containing more than 5% ambiguous nucleotides. Additionally, adapter sequences were removed from raw reads. Subsequently, the clean reads were de novo assembled into unigenes using Trinity (Grabherr et al., 2011). The statistical power of this experimental design, calculated in RNASeqpower (depth = 20, effect = 2, alpha = 0.05), yielded a power estimate exceeding 0.90. Hierarchical clustering analysis, implemented by Corset, was employed to generate the longest cluster sequences, which were subsequently identified as unigenes (Davidson & Oshlack, 2014). Gene expression analysis including transcript abundance estimation and normalization of expression values into FPKM (Fragments per kilobase of transcript per million fragments mapped), was conducted using the Trinity platform. Subsequently, a clustering heatmap was plotted using the R package Pheatmap. The differentially expressed genes (DEGs) were identified by DESeq2 (Liu et al., 2021) The Benjamini–Hochberg method was used to perform a multiple hypothesis test correction on the hypothesis test probability p-value < 0.05 to obtain the false discovery rate (FDR). DEGs were identified based on the thresholds of adjusted p-value < 0.05 and |log2Fold Change| > 1 (Love, Huber & Anders, 2014). All the DEGs underwent functional annotation and enrichment analyses. The annotation of gene functions was subjected to alignment (E-value < 10−5) using the following databases: Pfam (Pfam Protein families) (http://pfam.xfam.org/), Uniprot (Swiss-Prot), and KEGG (Kyoto Encyclopedia of Genes and Genomes database) (http://www.genome.jp/kegg). Gene Ontology (GO) and KEGG enrichment analyses with DEGs were performed using the GOseq R packages based on hypergeometric test (Young et al., 2010). Additionally, prediction of the TFs prediction was carried out using the iTAK v1.6 software package (Zheng et al., 2016). The Pearson correlation coefficient of the DEGs and DAMs was calculated using the cor function in R. The gene connection network was drawn with Cytoscape software version 3.10 (Su et al., 2014).

Quantitative real-time polymerase chain reaction (qRT-PCR) validation

The qRT-PCR analysis was performed using the Universal SYBR Green PCR Master Mix Kit (Beijing Lanjeke Technology Co., Ltd, Beijing, China.) on the Roche Lightcycler96 Fluorescent Quantitative PCR System (Jimenez et al., 2018). Fourteen genes exhibiting significant differential expression were chosen for validation via qRT-PCR analysis. The GAPDH gene served as an internal control (Wang et al., 2020).Specific primer pairs were designed using the Primer 5.0 software (Dossa et al., 2018), and the relative expression levels of the selected genes were normalized to the expression level of the ACTIN gene (Cao et al., 2017). All experiments were performed in triplicates, and the relative gene expressions were calculated using the 2−ΔΔCt method (Table S1) (VanGuilder, Vrana & Freeman, 2008). The primer sequences were listed in Table S2.

Results

Phenotypic characteristics and pigment content

With the exception of L* and a*, there are significant differences in other hue parameters (b*, C, and h°) were evident among the three species (Fig. 1B). CZ exhibited a significantly higher negative a* value in comparison to XB. The positive b* and C values exhibited a progressive increase in the order of XB<ZD<CZ with significant differences. Regarding the h° value, CZ approached closest approximation to 90°, followed by ZD, and XB (Fig. 1B, Table S3). These findings collectively indicate a gradual increase in petal yellowness across the species, with CZ displaying the highest level among the three species.

For all species, the total flavonoid content exhibited a statistically significant superiority over the total carotenoid content (P < 0.05). In addition, the total flavonoid and total carotenoid contents followed the order of XB<ZD<CZ, which aligned with the ranking of yellowness (b* value) among the three species (Fig. 1C and Table S4).

Differential metabolites identification

Across nine samples representing three distinct materials, a total of 356 flavonoid metabolites were identified and classified into 12 groups, including chalcones (16), flavanones (33), flavanonols (12), anthocyanidins (17), flavones (43), flavonols (131), flavonoid carbonosides (25), flavanols (22), tannins(31), biflavones (1), isoflavones (12), and proanthocyanidins (13).

Pairwise comparisons were conducted among three species (CZ vs. ZD, CZ vs. XB, and ZD vs. XB) to identify DAFs. A total of 221 DAFs (137 up-regulated and 84 down-regulated), 212 DAFs (116 up-regulated and 96 down-regulated), and 223 DAFs (108 up-regulated and 115 down-regulated) were identified in the CZ vs. ZD, CZ vs. XB, and ZD vs. XB comparisons, respectively (Fig. 2A). To elucidate the variations in flavonoid biosynthesis among the three materials, a KEGG pathway enrichment analysis of all DAFs was conducted. In the CZ vs. ZD comparison, the largest number of DAFs (12) exhibited significant enrichment in the flavone and flavonol biosynthesis pathway. Conversely, in the CZ vs. XB comparison, no metabolic pathways were significantly enriched; however, the flavonoid biosynthesis pathway demonstrated enrichment with the largest number of DAFs (16). Similarly, in the ZD vs. XB comparison, the flavonoid biosynthesis pathway exhibited the most significant enrichment, encompassing 17 DAFs (Fig. 2B). The substantial enrichment of DAFs indicates that flavonoids synthesis was the main factor contributing to the differences in flower color. A total of 36 DAFs were identified as shared across multiple pathways within flavonoid biosynthesis and its subbranches, including the flavonoid, flavone and flavonol, anthocyanin, and isoflavone biosynthesis pathways (Table S5). The cluster heat map revealed that 21 DAFs exhibited the significantly higher content in CZ compared to ZD and XB (Fig. 2C).

Figure 2 Differentially accumulated flavonoids (DAFs) of three species of Sect. Chrysantha Chang.

(A) The volcano plot of DAFs. Each point represents one metabolite, green represents down-regulated metabolites, red represents up-regulated metabolites, grey represents detected metabolites without significant change. (B) KEGG enrichment analysis of all DAFs. The color of the dot reflects the size of the p value. and the redder it is, the more significant the enrichment is. The size of the dot represents the amount of enriched metabolite. (C) Heat map clustering analysis of 36 DAFs mapped in the flavonoid pathways. Red and blue represented up-regulate and down-regulate.

Differentially expressed genes (DEGs) analysis

The sequencing of the nine sample libraries (three biological replicates for each of the three materials) yielded a range of 37.29 to 50.72 gigabases (Gb) raw reads. Following the removal of low-quality reads, clean reads accounted for 99.79% to 99.90% of the raw data. Quality assessments indicated Q20 and Q30 values of each library at 97.31% and 92.99%, respectively. The GC contents across samples ranged from 45.08% to 46.07% (Table S6). Finally, a total of 154,185 unigenes were retained (Table S7), out of which 47,277 were annotated (Table S8). Heatmap analysis uncovered consistent expression patterns among the three biological replicates of each material (Fig. S1).

Through pairwise comparisons among the three species (CZ vs. ZD, CZ vs. XB, and ZD vs. XB), a total of 12,948 DEGs were identified in CZ vs. ZD, comprising 6,559 up- regulated and 6,389 down-regulated genes. Similarly, CZ vs. XB exhibited 14,018 DEGs with 7,081 up- regulated and 6,937 down-regulated genes, while ZD vs. XB displayed 13,366 DEGs consisting of 6,669 up- and 6,697 down-regulated genes (p-value < 0.05, Fold change ≥ 2, Fig. 3A). The significantly enriched GO terms of DEGs were illustrated in the Fig. 3B, with “metabolic process” ranking among the top five in the biological process category. In addition, KEGG analysis highlighted significant enrichment of DEGs within four biosynthesis pathways, including the flavonoid biosynthesis pathway (37 DEGs), anthocyanin biosynthesis pathway (5 DEGs), isoflavonoid biosynthesis pathway (7 DEGs), and flavone and flavonol biosynthesis pathway (6 DEGs) (Fig. 3C). Further examination was conducted on the 37 shared DEGs enriched in the flavonoid biosynthesis pathways (Table S9).

Figure 3 Differentially expressed genes (DEGs) of three species of Sect. Chrysantha Chang.

(A) The volcano plot of DEGs, each point represents one gene, green represents down-regulated genes. red represents up-regulated genes, blue represents detected metabolites without significant change. (B) The GO enrichment scatter plot. (C) The KEGG enrichment scatter plot. Abscissa represents the degree of enrichment; ordinate represents enriched pathways; color of the point represents the g value, size of the point represents the number of enriched genes.

Transcription factors (TFs) identification

Transcription factors play a pivotal role as regulators in flavonoid biosynthesis, modulating gene expression. Based on the results of GO functional annotation (Fig. 4A and Table S10), a total of 157 differentially expressed TFs were identified, with 90 TFs identified in CZ vs. ZD, 99 in CZ vs. XB, and 115 in ZD vs. XB, respectively. The majority of these differentially expressed TFs were affiliated with five TF families, namely MYB, HD-ZIP, AP2/EREBP, WRKY, and MAD-box (Fig. 4B).

Figure 4 Identification of candidate transcription factors encoding key structural genes.

(A) The histogram of the number of TFs in each family. Red represents TFs families that participated in the correlation analysis. while blue did not. (B) The correlation analysis between differentially expressed TFs and key structural genes. Circles colored in “oink” meant the key structural genes. Triangles colored in “blue”mean TFs. Lines colored in “red and blue” represent positive and negative correlations.

Integrative analysis of transcriptome and metabolome

To identify pivotal metabolites, a correlation analysis between the DAFs and the flower color phenotype was conducted. The findings unveiled that ten metabolites exhibited a significant positive correlatation with yellowness (b* value), including comprising myricetin, quercetin, quercetin-3-O-rhamnoside, quercetin-3-O-sophoroside, quercetin-3-O-sambubioside, quercetin-3-O-rutinoside, luteolin-7-O-neohesperidoside and apigenin-7-O-glucoside, phloretin-2’-O-glucoside, and naringenin chalcone (|PCC| > 0.9, Fig. 5). To elucidate the regulatory impact of key structural genes on metabolites enriched in the flavonoid biosynthesis pathways, a correlation analysis was conducted between 36 DAFs and 37 DEGs (Table S11). The analysis identified eight DEGs that were significant associated with key metabolites (Fig. 6). The schematic depicts the potential involvement of the flavonoid pathway in CZ, ZD, and XB, suggesting their role in the accumulation of yellow pigment in petals (Fig. 7).

Figure 5 The correlation analysis of DAFs mapped in the flavonoid biosynthesis pathways and petal color phenotype.

Asterisks (***) represent a very significant correlation at the 0.001 level.

Figure 6 Heat map clustering analysis of 37 DEGs mapped in the flavonoid pathways.

Yellow and green represented up-regulated and down-regulated, respectively.

Figure 7 The core schematic diagram of avonoids synthesis in the Camellia section Chrysantha.

CHS, chalcone synthase; CHI, chalcone isomerase; F3H, flavanone 3-hydroxylase; F3’H, flavonoid 3’-monooxyfenase; F3’5’H, Flavonoid-3’5’-hydroxylase; FSⅡ, flavone synthase Ⅱ; DFR, dihydrofiavonol-4-reductase; ANS, anthocyanidin synthase; ANR, anthocyanidin reductase; LAR, leucoanthocyanidin reductase; 3GGT, anthocyanidin 3-O-glucoside 2’-O-glucosvltransferase ECGT; epicatechin, 1-O-galloyl-β-D-glucose O-galloyltransferase; Key genes and metabolites are highlighted in red.

To elucidate potential regulatory interactions, a Pearson correlation analysis was conducted between the expression levels of six key structural genes (CcCHI, CcFLS, CcDFR1, CcDFR2, CcDFR3, and CcCYP75B1) and the aforementioned five TFs families (Table S12). CcCHI and CcFLS positively contribute to pigment synthesis, thereby influencing yellow coloration. Conversely, the upregulation of CcCYP75B1 and CcDFR genes is associated with the accumulation of leucocyanidin and afzelechin. Additionally, Cc3GGT exhibited higher transcription, exhibiting a notable positive correlation with peonidin-3-O-glucoside. Meanwhile, the expressions of TFs were consistent with the contents of six above key flavonols. Within the MYB family, TFs CcMYB22 and CcMYB33 demonstrated a significant positive correlation with structural genes. TFs CcMYB6, CcMYB2, CcMYB38, and CcMYB35 provided negative regulation of flavonoid synthesis. In other families, high expression of CcHD-ZIP1, CcMADS-box1, and EREBP13 positively regulates structural genes, thereby promotes the accumulation of flavonoid pigments. Notably, among these TFs, CcMYB22, CcMYB28, CcMYB17, CcEREBP9, and CcEREBP13 exhibited high expression levels (FPKM ≥10).

Quantitative real-time polymerase chain reaction (qRT-PCR) analysis

To further check the reliability and accuracy of the transcriptome data, 14 genes were selected for qRT-PCR validation based on their significant differential expression. The qRT-PCR results were consistent with RNA-Seq data, with an R2 value of 0.82 (Fig. 8, Table S13). These findings validate the reliability and consistency of the transcriptome data utilized in this study.

Figure 8 Expression patterns of 14 flavonoid biosynthesis related differentially expressed genes (DEGs) by qRT-PCR.

Different letters above bars represent significant difference at P < 0.05.

Discussion

Flavonoid compounds affecting petal color in Sect. Chrysantha Chang

In the realm of botanical biology, flavonoids undertake diverse functions, including the regulation of cell growth, attraction of pollinating insects, and protecting against both biotic and abiotic stresses (Dias, Pinto & Silva, 2021). Acting as pigments within floral structures, flavonoids contribute to a broad array of hues, encompassing shades of red, purple, blue and yellow. The specific type and combination of flavonoids in a flower dictate its coloration (Deng, 2021). Our investigation involved a comprehensive quantitative and qualitive analysis of the petals from three species within Sect. Chrysantha Chang, characterized by a spectrum of flower colors ranging from yellow to milk-white. The result showed a correlation between the total flavonoid content and the degree of yellowness (b* value) in the petals of three species, aligning with the observed phenotypic variation in flower colors, namely, XB<ZD<CZ (Fig. 1). These findings strongly suggested the pivotal involvement of flavonoids in the formation of yellow colors among the studied species.

A total of 356 flavonoids were identified by metabolomic analysis, with 36 DFAs notably enriched in the well-known metabolic pathways. Furtherly, a Pearson correlation analysis between DAFs and yellowness (b* value) unveiled 10 significant positive correlations, comprising six flavonols, two flavones, and two chalcones (Fig. 4A). These DAFs were speculated to be closely associated with the deepening of the yellow color observed in the petals of golden Camellia. Specifically, the elevated levels of quercetin and luteolin, present as aglycones (quercetin, quercetin-3-O-rhamnoside, quercetin-3-O-sophoroside, quercetin-3-O-sambubioside, quercetin-3-O-rutinoside, and luteolin-7-O-neohesperidoside), were identified as major contributors (Table S5). Similarly, previous studies on C. nitidissim have demonstrated that various quercetin derivatives, including quercetin-7-O-glycoside (Qu7G), quercetin-3-O-glycoside (Qu3G), and quercetin, play a significant role in imparting yellow pigmentation of petals (Zhou, Li & Fan, 2012; Zhou et al., 2017). In addition, the potential influence of peonidin-3-O-glucoside on the vividness of petals observed in this study corresponds with findings from five reported species within Sect. Chrysantha Chang (Li et al., 2019).

Key structural genes involved in synthesis of key metabolites

The accumulation of flavonoids in plant tissues is intricately linked to multiple processes, including synthesis, degradation, and storage. In this study, a total of 154,185 genes were obtained by sequencing and assembly, following which 37 DEGs related to flavonoid synthesis were identified. Finally, eight key structural genes were determined to be involved in regulating the synthesis of key metabolites, including CcCHI(20522_c0_g1), CcCYP75B1(28055_c2_g1), CcFLS(22278_c3_g3), CcDFR1(22091_c0_g2), CcDFR2(27841_c2_g1), CcDFR3(30430_c0_ g1), CcLAR1(22577_c0_g2), and CcLAR2(25214_c1_g1). It has been reported that FLS, F3’H, and DFR genes exhibit complex interactions involving competition and cooperation. Changes in their expression profiles have been observed to significantly influence the synthesis of flavones, flavonols, and anthocyanidins (Schijlen et al., 2004).

Our correlation analysis showed that CcCHI positively regulated the synthesis of naringenin, thereby supplying more substrates for the subsequent production of key metabolites (Yamagishi, 2016). Furthermore, we found that CcFLS positively regulated the synthesis of flavonols, whereas CcDFR and CcLAR were positively associated with synthesis of flavanols. Conversely, CcDFR and CcCYP75B1 genes were found to negatively related the synthesis of flavones and flavonols. These findings indicated that the up-regulation of CcCHI and CcFLS promoted the synthesis of flavones and flavonols, thereby contributing to the intensification of yellow pigmentation in the petals. However, the up-regulation of CcLAR genes, which are associated with the synthesis of excessive colorless flavanols, may potentially diminishing the yellowness of the petals. In addition, CcDFR may influence flavanols synthesis under conditions of substrate competition. Based on these findings, we proposed a potential strategy for genetic engineering in breeding yellow Camellia, which involves up-regulation of CHI and FLS genes while simultaneously down-regulation of DFR and CYP75B1 genes. Moreover, the 3GGT gene was found to positively regulate peonidin-3-O-glucoside, which could potentially affecting the vividness of the petals (Li et al., 2019). Any changes within these pathways have the potential to affect the levels of flavonoid. These findings collectively emphasize that the formation of flower color involves a complex interplay of pigment synthesis and a range of intricate gene expression processes. However, further verification is warranted.

Candidate transcription factors regulating expression of key structural genes

In our study, a total of 157 differentially expressed TFs were identified in CZ, ZD, and XB, primarily attributed to five TF families (MYB, HD-ZIP, AP2/EREBP, WRKY, and MAD-box). Specifically, within these TFs, three candidates, CcMYB22(25744_c1_g3), CcMYB23(25259_c1_g3), and CcMYB33(22125_c2_g), were identified as potential promoters of flavonoid accumulation by regulating the CcFLS gene. Liu et al. (2022) aslo reported the positive correlation between FLS and flavonols accumulation in C. nitidssima. Additionally, CcMYB28(24527_c4_g1) may positively regulate the CcCHI gene. Conversely, CcMYB2(8054_c0_g1), CcMYB6(31451_c0_g2), CcMYB17(26771_c1_g1) and CcMYB38(17480_c0_g1) TFs may negatively regulate the CcDFR1, CcDFR2 and CcDFR3 gene. And CcMYB28(24527_c4_g1) and CcMYB35(20870_c0_g2) may act as repressors of the CcCYP75B1 gene. The CcMYB2, CcMYB6, CcMYB17, CcMYB28, CcMYB25, and CcMYB38 may inhibit regulation of DFR and CYP75B1, thereby contributing to the intensification of petal yellowness (Zhou et al., 2013). These TFs may function as repressors of flavonoid synthesis. The involvement of MYB and bHLH TFs in both the activation and repression of genes associated with pigment biosynthesis pathways has been documented in previous studies (Liu et al., 2016). Recent advancements in TF research have led to the discovery of new TF families associated with pigment synthesis (Yan et al., 2022; Chen et al., 2022). Our data indicated that CcHD-ZIP1, CcAP2, CcMYB28, CcEREBP9, EREBP13, and CcMADS-box1 effectively regulate the expression of multiple key structural genes. Five TFs with relatively high expression levels (FPKM > 10) were determined as the candidate TFs, namely CcMYB22, CcMYB28, CcMYB17, CcEREBP9, and CcEREBP13. These results suggest that these five candidate TFs and six key structural genes (CcCHI, CcFLS, CcDFR1, CcDFR2, CcDFR3, and CcCYP75B1) play a significant role in regulating flavonoid metabolites that influence petal color traits.

Conclusions

In this study, three species in Sect. Chrysantha Chang, each displaying a distinct series of yellow-toned petals, were collected to investigate their variation mechanisms. The metabolome analysis results indicated that flavonoid metabolites played a dominant role in determining the yellowness of the flowers. Ten metabolites, including myricetin, quercetin, quercetin-3-O-rhamnoside, quercetin-3-O-sophoroside, quercetin-3-O-sambubioside, quercetin-3-O-rutinoside, luteolin-7-O-neohesperidoside and apigenin-7-O-glucoside, phloretin-2’-O-glucoside, and naringenin chalcone, were closely related to the deepening of the yellowness. Transcriptome differential gene analysis revealed the involvement of six key structural genes and five candidate transcription factors in the regulation of flavonoid metabolites. Our findings offer a valuable insight in petals coloration mechanism of Sect. Chrysantha Chang, and provide candidate regulatory genes for breeding yellow Camellia varieties. In the future study, four regulatory expression genes (CHI, FLS, DFR and CYP75B1) need to be further validation, and to clarify their functions then using in transgenic breeding.

Supplemental Information

Supplemental Information 1 Heatmap of gene expression based on FPKM value of three species of Sect. Chrysantha Chang.

Supplemental Information 2 MIQE checklist for qRT-PCR.

Supplemental Information 3 Primers used in qRT-PCR.

Supplemental Information 4 Raw data for hue parameters of petal colour.

Supplemental Information 5 Raw data for carotenoid and flavonoid contents.

Supplemental Information 6 36 DAFs enriched in the flavonoid biosynthesis pathway of three species of Sect. Chrysantha Chang.

Supplemental Information 7 Summary of sequencing data and quality inspection.

Supplemental Information 8 The assembly results of transcriptome.

Supplemental Information 9 Numbers of unigenes annotated with various databases.

Supplemental Information 10 37 DEGs mapped in the flavonoid biosynthesis pathways of three species of Sect. Chrysantha Chang.

Supplemental Information 11 Transcription factors involved in correlation analysis.

Supplemental Information 12 The correlation analysis of DEGs and DAFs mapped in flavonoid metabolic pathways.

Supplemental Information 13 The correlation analysis between the key structural genes and TFs.

Supplemental Information 14 Raw data for qRT-PCR.

Additional Information and Declarations

Competing Interests

Author Contributions

Data Availability

The authors declare that they have no competing interests.

Yadan Yan conceived and designed the experiments, performed the experiments, authored or reviewed drafts of the article, and approved the final draft.

Ye Wang conceived and designed the experiments, performed the experiments, authored or reviewed drafts of the article, and approved the final draft.

Yafeng Wen conceived and designed the experiments, authored or reviewed drafts of the article, and approved the final draft.

Yu Huang conceived and designed the experiments, authored or reviewed drafts of the article, and approved the final draft.

Minhuan Zhang analyzed the data, prepared figures and/or tables, and approved the final draft.

Jiadi Huang analyzed the data, prepared figures and/or tables, and approved the final draft.

Xinyu Li analyzed the data, prepared figures and/or tables, and approved the final draft.

Chuncheng Wang analyzed the data, prepared figures and/or tables, and approved the final draft.

Dangqing Xu analyzed the data, prepared figures and/or tables, and approved the final draft.

The following information was supplied regarding data availability:

The original transcriptome sequencing data is available at NCBI: PRJNA1003846.

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
