# Peer review of "Metabolome and transcriptome integration reveals insights into petals coloration mechanism of three species in Sect. Chrysantha chang"

_PeerJ, doi:10.7717/peerj.17275_

## Round 0.1 · original submission · Major Revisions

Your manuscript was reviewed by three independent experts in the field. All the reviewers found the work interesting but raised several issues which should be addressed in revision. The reviewers provided detailed comments in their reviews and pointed out the areas where the manuscript needs to be improved.

**Language Note:** The review process has identified that the English language must be improved. PeerJ can provide language editing services - please contact us at [email protected] for pricing (be sure to provide your manuscript number and title). Alternatively, you should make your own arrangements to improve the language quality and provide details in your response letter. – PeerJ Staff

Reviewer 1 ·

Basic reporting

The research article explores the metabolic and transcriptional differences among the species of Sect. Chrysantha Chang with various degrees of yellow flowers. The article focuses on the TFs regulating the formation of yellow flowers in the Sect. Chrysantha Chang. The species that were selected were the Sect. Chrysantha Chang, the petal of C. chuongtsoensis (CZ) is golden yellow with a waxy luster, C. achrysantha (ZD) is light yellow, and C. parvipetala (XB) is milky white.
The research activity was divided into three parts which analyzed the types and contents of flavonoid metabolites and related gene expression in the petals at the blooming stage of different species. The study was aimed to (1) determine the main metabolites enhancing the yellowness of petals in Sect. Chrysantha Chang; (2) identify functional structural genes and TFs involved in flavonoid biosynthesis; and (3) explore the metabolic pathways and the mechanism of color formation in yellow Camellia petals.

The language used in the whole article is smooth and simple to understand the objective.
Literature references used against the research work are relevant. The tables and raw data supported the findings described in the research experiment model. The results that were presented were relevant.

Experimental design

The research question is very well defined and the intention of the research work is well taken. The experimental findings and data are also relevant. The investigation of the yellow and white color formation is very well studied and the mechanism is described with all the substances responsible;e for it.
The Research resulted in identification of key 10 metabolites and key structural genes which were significantly positively correlated with yellowness, including 6 flavonols (myricetin, quercetin, quercetin-3-O-rhamnoside, quercetin-3-O-sophoroside, quercetin-3-O-sambubioside, quercetin-3-O-rutinoside), 2 flavones (luteolin-7-O-neohesperidoside and apigenin-7-O-glucoside), and 2 chalcones (phloretin-2'-O-glucoside and naringenin chalcone).

Among the three samples that were taken up for an experiment, it was found that flavonoids were the pigment group determining petal color difference among the three materials. They also study that 8 Key structural genes involved in the synthesis of key metabolites involved in regulating the synthesis of key metabolites which regulate the mechanisms of flower color change.

They also focused their research on the possibility of other differentially expressed genes regulating flower color Auxin content in petals is significantly higher than in other plant organs and plays an important role in the morphological changes of flower organs. They have also laid down the prospects of further research on auxin signal transduction-related and transporter-related genes.

There are some corrections to be made -
Line numbers as per Word document

1. Line-117 – “1ml distilled water and 10 mg reagents” - Pl. elaborate the what kind of reagents were
used.
2. Line-126- 127 – "The resultant sample solution was configured, and then the absorbance (ΔA) was measured at 470nm" - It is mentioned that all the absorbance were taken at 440 nm, then why there is a deviation in this case, kindly explain.
3. Line-205 – "Indentification " - pl. correct the spell.
4. Line-268 – "[10-12]. " - what these numbers are referring to ??
5. Line-333 – "[Chen et al., 2023]. " - The year of publication written is wrong, it's 2021
6. Line-427 – "[He et al., 2022] " - Pl. check the year of publication
7. Line -524-526 - reference - 20, the year of publication is 2014 -https://doi.org/10.1111/jph.12354.
8. Line-588-589 - Pl. mention full details of the paper - Plants 2021, 10(10), 2065; https://doi.org/10.3390/plants10102065 .

9. The supplemental file - MIQE checklist for RT-qPCR. - is not opening - corrupted it seems. Pl. reload again.
10.

Validity of the findings

The article focuses on the flavonoids and various transcription factors regulating the expression of key structural genes responsible for color formation in fruits and flowers.
The research references cited are relevant and are sound.

conclusion of the research activity is well stated and further research activity is very well defined.

Additional comments

NA

·

Basic reporting

Dear authors
The following modifications are required
Abstract
 In general, this section is poorly written. It is written simply. This section should include. As a result, this section should be improved.
 Before describing the goal, the authors must define the issue in a single line and explain why they chose this approach to study this review.
 Some scored data should be added
 In the final line of the abstract, the authors should present a decisive conclusion derived from the research and provide a single line of future prospects.
Keywords
 The content of keywords did not reflect the content of this manuscript and the words used for forming the title should not be used as the keywords. So, the structure of keywords should be changed.
Introduction
 The authors should give some lines about the knowledge gap which their reviews have covered along with the hypothesis statement
 Also, the authors should provide a novelty statement at the end. What new things authors have done or correlated in this research compared to old ones?
 The general and specific aim should be specified
Materials and Methods
 All procedures should be supported by the references
 The authors should write the number of replications and the number of plants per replications.
 The authors should mention the type of tissue used for the biochemical tests.
 All abbreviations should be written in full name
 The statistical data analysis should be included
 The method of comparison of means (LSD, Tukey…etc) should be added

Results and discussion
 In general, the figures are not presented clearly.
 The status of significant of each studied parameter should be mentioned at the beginning of the text
 The authors should mention the scored data when they explain the maximum and minimum of the studied traits.
 All captions should be improved, showing the contents of tables and figures
 The method of the comparison of means (LSD, Duncan, Tukey, Dunnett) should be included
 The discussion is weak. The authors should interpret all results obtained in this study by adding some information about the results obtained in their study. The authors should explain how all of the findings from this study relate to their own findings.
Conclusion
 The authors should summarize the most significant findings because they have written this section in an easy-to-read manner.
 Future works about this research should also include additional works

Experimental design

The method of comparison between the means is not available

Validity of the findings

It needs some improvement

Reviewer 3 ·

Basic reporting

This study is interesting and comprehensively examined metabolic and transcriptomic patterns of three species in Sect. Chrysantha Chang.

Experimental design

The experimental design and the statistical analysis were valuably carried out.

Validity of the findings

They have detected flavonoids and studied gene expression in the respective species. However, Syber green and delta delta ct is less accurate in comparison with standard curves and Taqman method. Despite observing acceptable data, there is no novelty in the main findings.

Additional comments

Thank you very much

---

## Round 0.2 · Minor Revisions

Although the authors have improved the manuscript significantly, however manuscript still needs a revision as suggested below.

Abstract: Authors should expand the abbreviation of species name of CZ, ZD and XB.

Ln 66: "Camellia is a famous ornamental plant" this is wrong, Camellia is not single plant species and is a genus. Authors should correct this.

There are a lot of typographical errors throughout the manuscript as I can see in the manuscript word file.

**Language Note:** The Academic Editor has identified that the English language must be improved. PeerJ can provide language editing services - please contact us at [email protected] for pricing (be sure to provide your manuscript number and title). Alternatively, you should make your own arrangements to improve the language quality and provide details in your response letter. – PeerJ Staff

Reviewer 1 ·

Basic reporting

Queries raised were well addressed

Experimental design

No comments

Validity of the findings

No comments

Additional comments

No comments

---

## Round 0.3 · Major Revisions

Your paper was evaluated by the editorial board, and it was found that the English language, particularly the abstract, requires further improvement, with nearly every sentence carrying a grammatical error. A few examples are provided below.
line 23 "a...ornamental plant". We aren't talking about one plant but several species. line 33 "annotate to the KEGG pathway". There isn't a "KEGG pathway". There is a KEGG pathway database. We speculated 10" should be "We speculated that 10". Line 33 "Transcriptome analysis indicated that it had" ...should be "...analysis indicated that there were". line 34 "Six key genes..." this is not a complete sentence.

The editorial board also raises a number of other questions:
line 180. the delta-delta CT method is a qRT-PCR quantification method; why is it listed in this section on RNAseq?
lines 181-184. It is incorrect to input FPKM into DEseq. Raw reads should be used.
line 185: the procedures for functional annotation need to be described.
Multiple testing correction (FDR) should be applied to the correlation tests
Figure 2, 5, 6: text is too small to read.
The assembled transcripts should be provided either as a zipped fasta file (supplement) or uploaded to NCBI."

Therefore, I would suggest to authors to revise the manuscript thoroughly particularly the English language.

**Language Note:** The Academic Editor has identified that the English language must be improved. PeerJ can provide language editing services - please contact us at [email protected] for pricing (be sure to provide your manuscript number and title). Alternatively, you should make your own arrangements to improve the language quality and provide details in your response letter. – PeerJ Staff

---

## Round 0.4 · Minor Revisions

Though the current version is improved however it still has several typographical errors. Authors need to check them thoroughly throughout the manuscript. Few are mentioned below;
Ln 50: Delete the repeated word "diverse"
Ln 302: "signifcant" to "significant", "diferential" to "differential"
Ln 308: "influenc" to "influence"
Ln 392: "investigat" to "investigate"
Ln 286: "conduced" to "conducted"
Ln 158: "base" to "based"
Ln 121-123: This sentence is not clear. "To determine the total carotenoid content, 0.1 g of fresh petals from each species was mixed with 1 ml of distilled water and a 10 mg aqueous acetone solution". What is 10 mg aqueous acetone solution? The solution can not be in mg.

---

## Round 0.5 · Minor Revisions

I am still not convinced with the below statement.
"To determine the total carotenoid content, 0.1 g of fresh petals from each species was mixed with 1 ml of distilled water and a 10 mg of acetone."
What do you mean by "1 ml of distilled water and a 10 mg of acetone"? Given that acetone is a liquid solvent, then how so low quantity (10 mg) of acetone was measured that too in the weight instead of volume. Please clarify and change the statement.

---

## Round 0.6 · Minor Revisions

In your method you mentioned that "10 mg of reagent 1 (provided in the kit)". In order to improve the reproducibility, the authors must provide the exact detail of the kit including catalogue number and manufacture's details. If authors followed any published protocol they may cite that in the manuscript.

---

## Round 0.7 · accepted · Accept

Authors have addressed all the concerns raised during the review process. Therefore, manuscript is ready for publication.